# Adherence to Antihypertensive Therapy: A Cross-Sectional Study Among Patients in the Republic of Kazakhstan

**DOI:** 10.3390/ijerph22101483

**Published:** 2025-09-25

**Authors:** Akbayan Markabayeva, Aiman Kerimkulova, Riza Nurpeissova, Gyulnar Zhussupova, Ayagyoz Umbetzhanova, Dinara Zhunussova, Alisher Idrisov, Ardak Zhumagaliyeva, Aliya Seidullayeva, Aigul Utegenova, Lyudmila Pivina

**Affiliations:** 1Department of Family Medicine No. 2, NJSC “Astana Medical University”, Beibitshilik Str. 49a, Astana 010000, Kazakhstan; k-aiman@yandex.ru (A.K.); nurpeissova.r@amu.kz (R.N.); aliser73@mail.ru (A.I.); zhumar_77@mail.ru (A.Z.); 2Heart Rhythm Research Institute, NJSC “Astana Medical University”, Beibitshilik Str. 49a, Astana 010000, Kazakhstan; gulnar.zh1@gmail.com; 3Department of General Practice with the Course of Evidence-Based Medicine, NJSC “Astana Medical University”, Beibitshilik Str. 49a, Astana 010000, Kazakhstan; umbetzhanova.a@amu.kz (A.U.); dinarazhunussova7@gmail.com (D.Z.); 4Department of Pediatric Infectious Diseases, NJSC “Astana Medical University”, Beibitshilik Str. 49a, Astana 010000, Kazakhstan; seidullayeva.aliya@gmail.com; 5Department of Microbiology and Virology, NJSC “Astana Medical University”, Beibitshilik Str. 49a, Astana 010000, Kazakhstan; utegenova.a@amu.kz; 6Department of Emergency Medicine, NJSC “Semey Medical University”, Abay Str. 103, Semey 071400, Kazakhstan; semskluda@rambler.ru

**Keywords:** arterial hypertension, medication adherence, MMAS-8, Kazakhstan, non-communicable diseases, public health policy

## Abstract

**Background:** Poor adherence to antihypertensive therapy is a major barrier to effective blood pressure control, particularly in countries with a high burden of non-communicable diseases. In Kazakhstan, improving adherence is a key objective of the “Densaulyk” State Health Program (2020–2025). **Objective:** To assess medication adherence among patients with arterial hypertension in Kazakhstan and identify associated socio-demographic and clinical factors. **Methods:** A cross-sectional survey was conducted among outpatient hypertensive patients at a major urban medical center. Adherence was measured using the 8-item Morisky Medication Adherence Scale (MMAS-8). Socio-demographic characteristics, disease duration, and the number of prescribed medications were analyzed in relation to adherence levels. **Results:** Adherence was significantly associated with age, ethnicity, education, marital and financial status, disease duration, and treatment complexity. A notable share of participants demonstrated low to moderate adherence. The use of self-reported data may have introduced bias. **Conclusions:** Medication adherence in Kazakhstan is influenced by multiple interrelated factors. Targeted and culturally appropriate interventions—such as simplified regimens, digital tools, and broader access to subsidized drugs—are essential to improve long-term outcomes in hypertension management.

## 1. Introduction

Arterial hypertension (AH) remains one of the leading causes of premature mortality worldwide. According to the World Health Organization, more than 1.28 billion individuals aged 30 to 79 years suffer from elevated blood pressure, with approximately 46% of them unaware of their condition [1]. AH significantly increases the risk of cardiovascular diseases, including myocardial infarction, stroke, heart failure, and chronic kidney disease [2]. Consequently, it is a major contributor to the global burden of disease, and its timely detection and management are critical priorities for healthcare systems across most countries [3,4].

Findings from the Global Burden of Disease (GBD) 2019 study identified elevated blood pressure as the leading risk factor for mortality, contributing to over 10 million deaths annually [5]. Despite the availability of effective pharmacological treatments and clinical guidelines, only about 20% of patients in low- and middle-income countries achieve target blood pressure levels [6]. Despite a global effort to combat hypertension, adherence rates in low- and middle-income countries (LMICs) remain highly variable, often falling below 50%. Recent analyses suggest that cultural, structural, and economic barriers contribute to these low rates [7,8].

Cardiovascular diseases, including arterial hypertension (AH), are the leading cause of mortality in Kazakhstan. Each year, more than 40,000 Kazakhstani citizens die from these conditions, accounting for approximately 84% of total mortality in the country [9]. One of the pressing issues in current cardiology practice in the Republic of Kazakhstan is the low level of patient adherence to antihypertensive pharmacotherapy. Despite the availability of diagnostic and treatment options, only about 20% of patients achieve target blood pressure levels, a figure significantly lower than that observed in high-income countries [6]. In Kazakhstan, this problem is particularly acute, with local studies reporting poor adherence despite established treatment guidelines and drug availability [9,10].

A major challenge in the management of AH remains poor adherence to prescribed medication regimens. A recent systematic review encompassing over 100 studies reported that average adherence rates range from 40% to 60%. The most frequently cited reasons for nonadherence include forgetfulness, fear of side effects, distrust in medications, and the complexity of treatment regimens [11]. Furthermore, nonadherence can be categorized as either intentional or unintentional: the former is typically driven by patient beliefs, while the latter often results from forgetfulness or logistical barriers [12].

The World Health Organization (WHO) classifies factors influencing medication adherence into five categories: socio-economic factors, healthcare system-related factors, disease-related factors, therapy-related factors, and patient-related factors [13]. The most vulnerable population group remains older adults, particularly those over the age of 60, among whom adherence to treatment regimens significantly declines [14].

Various strategies have been employed to improve medication adherence, including simplification of treatment regimens (e.g., fixed-dose combinations), enhanced pharmaceutical care, mobile applications, reminder systems, and electronic prescriptions [15,16]. Digital monitoring tools, such as mobile health apps and smartwatches, facilitate tracking of medication intake and can alert healthcare providers in cases of nonadherence [17].

Pharmaceutical counseling has also shown high efficacy. According to randomized controlled trials, the inclusion of pharmacists in the care team for patients with hypertension leads to improved blood pressure control and increased adherence rates [18]. Furthermore, such approaches help identify individual barriers and provide personalized support [19].

Thus, the chronic nature of AH necessitates a comprehensive approach to improving treatment adherence, incorporating digital technologies, multidisciplinary care teams, and educational interventions. These strategies should be adapted to the socio-cultural context of the country and their effectiveness should be systematically evaluated in clinical practice.

## 2. Materials and Methods

This prospective cross-sectional study was conducted from March 2024 to September 2024 in four regions of the Republic of Kazakhstan: Semey, Astana, Aktobe, and Kokshetau. The study was implemented across five primary healthcare centers (PHCs), including four urban centers and one rural center located near one of the cities.

The research protocol entitled “Comprehensive Assessment and Improvement of the Organization of Primary Healthcare for Patients with Arterial Hypertension” was approved by the independent local ethics committee of the State Institution “Astana Medical University” (Decision No. 2 dated 10 November 2022). All study participants provided written informed consent prior to enrollment.

### 2.1. Participants

The study included adult patients (≥18 years) with a confirmed diagnosis of essential arterial hypertension. A total of 1125 respondents were enrolled in the sample. Patients were selected using a random sampling method from those who presented to the designated primary healthcare centers. Exclusion criteria comprised secondary arterial hypertension, decompensated endocrine or oncological diseases, pregnancy, severe acute or recurrent illnesses requiring hospitalization or intensive care, and the presence of advanced heart or respiratory failure.

### 2.2. Data Collection

Data were collected through face-to-face interviews using a standardized questionnaire. The questionnaire captured sociodemographic characteristics (age, sex, educational level, marital status, employment status, and socioeconomic status), behavioral factors (smoking, alcohol consumption), and the presence of comorbid conditions.

### 2.3. Assessment of Medication Adherence

Adherence to antihypertensive therapy was assessed using the Morisky Medication Adherence Scale (MMAS-8). For analysis purposes, adherence levels were categorized dichotomously into adherent and non-adherent groups. The level of adherence among participants was evaluated using the Morisky Widget MMAS-8 software. The MMAS-8 is a validated and reliable self-report tool widely used to assess medication adherence in patients with chronic conditions such as hypertension. The scale comprises eight items: the first seven are yes/no questions, and the final item uses a five-point Likert scale [20,21,22,23,24].

Do you sometimes forget to take your hypertension medication?People sometimes miss taking their medications for reasons other than forgetting. Thinking over the past two weeks, were there any days when you did not take your hypertension medication?Have you ever cut back or stopped taking your hypertension medication without telling your doctor because you felt worse when you took it?When you travel or leave home, do you sometimes forget to bring along your hypertension medication?Did you take your hypertension medication yesterday?When you feel your symptoms are under control, do you sometimes stop taking your hypertension medication?Taking medication every day is a real inconvenience for some people. Do you ever feel hassled about sticking to your hypertension treatment plan?How often do you have difficulty remembering to take all your hypertension medications?

Responses to the questionnaire were coded using the Morisky Widget MMAS-8 software, and results were scored accordingly. In this study, patients were classified into two groups-adherent and non-adherent—using a six-point cutoff [25]. Following the completion of the questionnaire, patients who scored less than 6 points were informed that they were classified as non-adherent. Patients with scores greater than 6 points were considered adherent. Score differences equal to zero were excluded from the analysis of non-adherence type classification.

### 2.4. Statistical Analysis

Continuous variables were presented as means (M) and standard deviations (SD). Bivariate associations were assessed using Pearson’s chi-square test. Associations between independent variables and non-adherence were analyzed using multivariable logistic regression. Both crude and adjusted odds ratios (ORs) with 95% confidence intervals (CIs) were calculated. The reference category for each variable was selected based on the most favorable profile.

To obtain the final regression model, the backward elimination method was applied. This approach is preferred over forward selection due to a lower risk of Type II error and greater statistical power compared to forced entry methods for the given sample size [26]. All analyses were conducted using the Statistical Package for Social Sciences (SPSS), version 23.0 (SPSS Inc., Chicago, IL, USA).

## 3. Results

A total of 1125 respondents were included in the study. The mean age of participants was 60.1 years (SD = 10.6). The majority were women (56.9%, *N* = 640), and 71.1% (*N* = 800) identified as ethnically Kazakh. Regarding place of residence, 91% (*N* = 1024) of respondents were urban dwellers. Nearly half of the participants (43.2%, *N* = 486) had completed higher education. A total of 48.8% (*N* = 549) were retired and not employed. Most respondents were married (77.2%, *N* = 869). The majority of patients were non-smokers (51.2%, *N* = 576). Overweight status was observed in 42.8% (*N* = 482) of participants.

Approximately 28.4% (*N* = 319) of all patients had coexisting cardiovascular diseases, while ischemic heart disease ranked second in prevalence at 19.5% (*N* = 219), followed by gastrointestinal disorders (18.6%, *N* = 209). Additionally, around 13% (*N* = 140) of patients reported either eye diseases or respiratory conditions. Notable proportions of other comorbidities included respiratory diseases (10.5%, *N* = 118), angina (9.8%, *N* = 110), neurological disorders (6.9%, *N* = 78), and kidney diseases (6.9%, *N* = 78).

Overall, **91.5%** (*N* = 1029) of respondents were classified as **non-adherent** to antihypertensive medication therapy according to the Morisky Widget MMAS-8 software. A detailed description of the cohorts is presented in Table 1.

Table 1 presents the distribution of demographic, socioeconomic, and clinical characteristics of patients with arterial hypertension according to their adherence to prescribed therapy. In this study, participants were categorized into adherent and non-adherent groups, followed by a comparative analysis. The mean age was 58.7 ± 11.2 years in the adherent group and 60.3 ± 10.6 years in the non-adherent group; however, this difference was not statistically significant (*p* = 0.24).

A significant difference was found in adherence by ethnicity (*p* = 0.001), with multivariable analysis indicating that Kazakh ethnicity was associated with a lower likelihood of adherence (aOR = 0.47). This finding highlights the need for culturally tailored interventions to address barriers specific to this group. As most detailed characteristics are shown in Table 1, only key statistically significant findings are summarized here.

Educational level also had a significant impact on treatment adherence (*p* = 0.002). Individuals with higher education were predominant in the adherent group (58.3%), whereas the non-adherent group had a greater proportion of patients with secondary and vocational education. Place of residence (urban vs. rural) also differed significantly between groups (*p* = 0.03), which may be attributed to better access to healthcare resources in urban areas.

Marital status showed a statistically significant association with adherence (*p* = 0.003). The majority of adherent patients were married (72.9%), while the non-adherent group included a higher proportion of single and divorced individuals.

Duration of hypertension was also correlated with adherence level (*p* = 0.03). While patients with a shorter disease duration (<1 year and 1–5 years) represented a significant proportion in both groups, the adherent group included a larger share of patients with over 20 years of hypertension (15.6%), possibly reflecting accumulated experience and a greater understanding of the importance of therapy.

Financial status, measured by self-reported satisfaction, showed a significant difference between groups (*p* = 0.001). Patients with high financial satisfaction (8–10 points) were more commonly observed among the adherent group (40.6%), whereas moderate satisfaction was more prevalent among non-adherent patients.

Body Mass Index (BMI) was also associated with adherence (*p* = 0.02); patients with a normal weight were more likely to be adherent, whereas those with excess weight were more frequently non-adherent. Smoking status emerged as a significant factor (*p* < 0.001), with a higher proportion of smokers in the non-adherent group.

The presence of comorbid conditions—including cardiovascular diseases, neurological disorders, kidney diseases, and others—also differed significantly between the groups (*p* < 0.05). Adherent patients more frequently reported having comorbidities, which may indicate increased motivation to follow prescribed therapy in the context of complications.

The number of prescribed medications per day demonstrated statistically significant differences (*p* < 0.001); a higher proportion of adherent patients were taking a single medication, whereas non-adherent individuals more commonly took two or more medications.

Figure 1 presents the results of a multivariable logistic regression analysis aimed at identifying factors associated with non-adherence to antihypertensive therapy. The following variables were assessed: ethnicity, education level, marital status, financial satisfaction, duration of hypertension, and number of prescribed medications. Ethnicity was also significantly associated with adherence. Respondents of Kazakh ethnicity had a lower likelihood of being adherent compared to individuals of other ethnic backgrounds (aOR = 0.43; 95% CI: 0.24–0.79).

In terms of education, respondents with incomplete secondary education showed a slightly higher likelihood of non-adherence (aOR = 1.71; 95% CI: 0.15–3.41) compared to those with higher education. Similarly, individuals with complete secondary education and vocational training demonstrated elevated odds of non-adherence (aOR = 1.76; 95% CI: 0.92–3.35 and aOR = 1.48; 95% CI: 0.82–2.69, respectively).

Patients who were married did not differ in their likelihood of non-adherence compared to unmarried individuals (aOR = 1.00). However, single patients demonstrated a higher likelihood of non-adherence (aOR = 0.75; 95% CI: 0.26–2.10), which may be related to the absence of social support. Divorced and widowed participants did not exhibit significant differences in adherence levels compared to married individuals (aOR = 1.35 and aOR = 0.86, respectively).

Patients with a low level of satisfaction with their financial status were significantly more likely to be non-adherent (aOR = 2.46; 95% CI: 0.99–6.07), potentially due to economic barriers affecting access to medical services. Those with moderate financial satisfaction also showed increased odds of non-adherence (aOR = 1.95; 95% CI: 1.24–3.07), further highlighting the influence of financial factors on treatment behavior.

With respect to duration of hypertension, patients with 1–5 years and 6–10 years of disease duration showed moderately higher odds of non-adherence (aOR = 1.23 and aOR = 1.55, respectively) compared to those with hypertension lasting more than 20 years.

Regarding medication burden, patients taking a single medication had a significantly lower likelihood of non-adherence (aOR = 0.11; 95% CI: 0.05–0.28). Among those taking two medications, the likelihood of non-adherence increased (aOR = 0.48; 95% CI: 0.27–0.84). However, for patients taking three or more medications, the risk of non-adherence increased again, though the effect was less pronounced (aOR = 0.77; 95% CI: 0.43–1.40).

## 4. Discussion

The present study revealed an alarmingly low level of adherence to antihypertensive therapy among patients in Kazakhstan: only 8.5% of respondents were classified as adherent according to the MMAS-8 scale. The low adherence rate identified in this study may initially appear unexpected; however, it reflects patterns commonly reported in similar settings. In middle-income countries, factors such as insufficient patient education, high out-of-pocket expenses for medications, and irregular clinical follow-up often contribute to reduced adherence. Additionally, the MMAS-8 instrument applies rigorous criteria for classification, which may result in lower rates of adherence when compared to less stringent tools. These aspects should be taken into account when interpreting the findings. This underscores the need for comprehensive interventions aimed at improving adherence, particularly in urban outpatient populations. This figure is significantly lower than global averages and highlights the urgent need for the implementation of comprehensive interventions that address not only clinical factors but also socioeconomic, educational, and cultural dimensions influencing treatment adherence.

A meta-analysis has shown that nonadherence to antihypertensive medication may be more prevalent among African and Asian populations, with an estimated prevalence of approximately 62.5% [27]. In high-income countries such as the United States and Germany, adherence rates range from 70% to 85% [28]. A study conducted in the United Kingdom reported that around 50% of patients fail to take their medications regularly, despite broad access to healthcare services and a high standard of care. Even in transitioning economies such as Brazil and India, adherence rates have been reported at 30–40% [29,30].

Thus, the findings from our study indicate that the level of adherence in Kazakhstan is substantially lower, even when compared to other resource-limited settings. Our results are consistent with those of the large cohort study PREDIcT-HTN conducted in Bangladesh, where only 7.2% of patients demonstrated good adherence to therapy [31]. Despite differences in assessment methods (Hill-Bone scale vs. MMAS-8), both studies underscore the persistent challenge of poor adherence to antihypertensive treatment in countries with developing healthcare systems.

Socio-demographic characteristics were also found to be significant determinants of treatment adherence. Adherence levels increased with age, particularly among those aged 60 years and older—a trend also observed in the study by Mills et al. [32] and confirmed in the PREDIcT-HTN cohort [31]. Older patients are likely more aware of the risks associated with hypertension and more disciplined in following treatment regimens. In contrast, younger patients (under 40 years) exhibited lower adherence, possibly due to an underestimation of the seriousness of the disease [33].

Our study also identified a significant association between educational level and adherence. Patients with higher education were more likely to be adherent, consistent with findings from studies in China, Turkey, and Bangladesh [30,34,35]. Higher educational attainment is typically associated with a better understanding of the importance of treatment and greater capacity for self-management.

Financial status played an equally important role. Patients satisfied with their financial situation demonstrated better adherence, whereas those with low income were more likely to miss doses. This aligns with findings from South Asia and Africa, where limited financial capacity is strongly associated with poor adherence to therapy [36,37]. A key confirmation of the financial barrier in Kazakhstan is provided by the study of Kozhekenova et al. [10], which investigated quality of life in patients with heart failure. The authors reported that the greatest deterioration occurred in the social and emotional domains, where many patients cited lack of financial means to purchase medications as the primary barrier. These findings underscore a systemic issue of economic accessibility to treatment.

In the present study, a significant association was identified between the duration of arterial hypertension and adherence to treatment. Patients who had been diagnosed with hypertension for more than five years demonstrated significantly higher levels of adherence compared to those who were newly diagnosed.

This finding is supported by two previous studies that identified disease duration as an independent predictor of medication adherence. Patients with over 10 years of disease history were significantly more likely to adhere to their treatment regimen compared to those with a diagnosis of less than five years (aOR = 1.598; 95% CI: 1.115–2.291 and aOR = 1.93; 95% CI: 1.07–2.92, respectively) [10,12].Clinical characteristics, including the number of medications prescribed, also influenced adherence. Several studies have demonstrated that the likelihood of treatment adherence is affected by the number of medications a patient is currently taking. Jankowska-Polańska et al. reported that individuals prescribed monotherapy (aOR = 1.67; 95% CI: 1.18–2.33) or single-pill combination therapy (aOR = 3.70; 95% CI: 1.56–8.33) were significantly more likely to adhere to treatment regimens compared to those receiving multiple separate medications [38]. Similarly, findings from Teshome et al. [37] showed that patients taking fewer than two antihypertensive medications per day were significantly more likely to adhere to treatment (aOR = 3.04; 95% CI: 1.53–6.06) compared to those on two or more medications. Additional studies from Poland and South Korea have also confirmed that patients receiving monotherapy exhibited better adherence than those taking three or more medications [38]. 

Ethnicity also emerged as a significant factor. Patients of Kazakh ethnicity demonstrated a lower likelihood of adherence to antihypertensive therapy, as shown in the multivariable regression model (aOR = 0.43; 95% CI: 0.24–0.79). While this finding contrasts with descriptive statistics, it suggests that other confounding factors may influence observed adherence patterns in this group. Cultural norms, language-related barriers, or health literacy gaps may partially explain this association and should be explored in future research. Potential explanations may include cultural characteristics, stronger family support systems, or better access to health information in the native language. However, this area warrants further sociological investigation to clarify underlying mechanisms.

Our findings are consistent with previous research conducted in Kazakhstan. The level of adherence, as measured by the MMAS-8 scale, was 5.16 ± 3.36 (95% CI: 4.79–5.52) in the experimental group and 5.18 ± 3.32 (95% CI: 4.82–5.55) in the control group, indicating that adherence levels were similarly low in both groups [39]. These results confirm that low adherence is a persistent trend across different regions of the country. 

This study focused on the issue of treatment adherence, which is often overlooked when analyzing factors affecting the effectiveness of hypertension therapy. Identifying high-risk patient groups susceptible to both poor blood pressure control and non-compliance with treatment regimens allowed us to focus on developing more targeted preventive and therapeutic measures. The large sample size ensured that the results were highly representative. Given that respondents were selected in a large specialized medical institution covering broad socio-demographic strata of the population and serving regions with diverse economic conditions, the validity of the study findings is significantly strengthened. The additional value of the work is provided by the careful collection of data on the social, demographic, and medical characteristics of the participants, as well as the use of analysis methods that take into account mixed effects, which made it possible to increase the reliability of the interpretation of the results. It should be noted that the study was based on a questionnaire, where the information was provided by the participants themselves, which is typical for cross-sectional studies. Despite possible biases associated with subjectivity and memory effects, a validated self-report scale was used to assess adherence, and the level of patient knowledge was assessed based on criteria verified by two independent family medicine specialists. 

The study was conducted in one of the largest medical centers in Kazakhstan, located in the capital, Astana, which allowed us to cover a wide range of outpatients. However, due to the design and sampling method, generalization of the results to the entire population requires caution, and causal relationships between risk factors and adherence levels cannot be established with complete confidence. Despite these limitations, the study provides valuable information on the current state of adherence to antihypertensive therapy and its determinants among outpatients.

Taking into account the implementation of the State Program for Healthcare Development of the Republic of Kazakhstan “Densaulyk” for 2020–2025, aimed at reducing mortality from non-communicable diseases, improving adherence to treatment should be one of the key areas. To achieve this goal, it is advisable to implement educational initiatives aimed at the young population and groups with low levels of health literacy; expand access to subsidized drugs; simplify treatment regimens using fixed-dose combinations; use digital technologies (e.g. reminders and mobile applications); regular monitoring of adherence in the primary health care system; and develop partnerships between health care institutions, social services, pharmacy chains, and local authorities [39]. Thus, the problem of insufficient adherence requires a comprehensive, intersectoral approach with an emphasis on the sustainable development of Kazakhstan’s public health system.

Beyond individual-level factors, structural health system barriers—such as limited availability of antihypertensive medications in primary care settings, inconsistent drug supply chains, and insufficient counseling services - likely contribute to poor adherence in Kazakhstan. Similar barriers have been documented in LMICs across Central Asia and Eastern Europe [39].

Cultural perceptions about chronic illness, traditional remedies, and reliance on acute care services rather than preventive care may shape patients’ adherence behaviors in Kazakhstan. For example, mistrust in long-term pharmacotherapy or fear of medication dependence can lead to intentional nonadherence, particularly in older adults or rural populations [40].

Some predictors in the logistic regression model had wide confidence intervals, indicating limited statistical precision. This may be due to small subgroup sizes or residual confounding. We acknowledge this as a limitation of the current analysis [41].

Another important determinant of medication adherence that was not included in our analysis is the presence of medication-related adverse effects. Numerous studies have demonstrated that side effects—such as dizziness, fatigue, cough, or sexual dysfunction—may lead patients to intentionally skip or discontinue antihypertensive therapy. Although these data were not collected in the current study, this limitation should be acknowledged. Future research would benefit from incorporating patient-reported outcomes related to drug tolerability, which may help to identify modifiable contributors to non-adherence and improve individualized treatment strategies.

## 5. Study Limitations

Despite the significant results obtained, this study has a number of limitations that must be taken into account when interpreting the findings. 

Additionally, certain clinically relevant comorbidities (e.g., diabetes, dyslipidemia, myocardial infarction) were not included in the analysis due to inconsistent documentation in the dataset.

Despite the widespread use of the MMAS-8 in clinical and epidemiological studies, this tool remains subjective and is not able to fully replace objective assessment methods, such as analysis of pharmacy data on drug receipt or electronic monitoring of drug intake.

Self-reported adherence may be affected by recall bias and social desirability bias, potentially leading to overestimation of adherence rates.

Additionally, as the majority of participants were recruited from urban healthcare settings, the results may not be fully generalizable to rural populations, where healthcare access and patient behavior may differ significantly.

Nevertheless, the presented study provides valuable empirical data on the nature of adherence to treatment among patients with arterial hypertension in the Republic of Kazakhstan. It highlights the importance of socio-demographic and economic factors influencing adherence to prescribed therapy and can serve as a basis for developing targeted strategies within the framework of public health policy.

## 6. Conclusions

The analysis revealed that non-adherence to antihypertensive treatment is influenced by multiple factors, including age, ethnicity, educational level, marital status, financial status, duration of the disease, and the number of prescribed medications. These factors can serve as the basis for developing targeted interventions and improving treatment strategies aimed at enhancing patient adherence.

## Figures and Tables

**Figure 1 ijerph-22-01483-f001:**
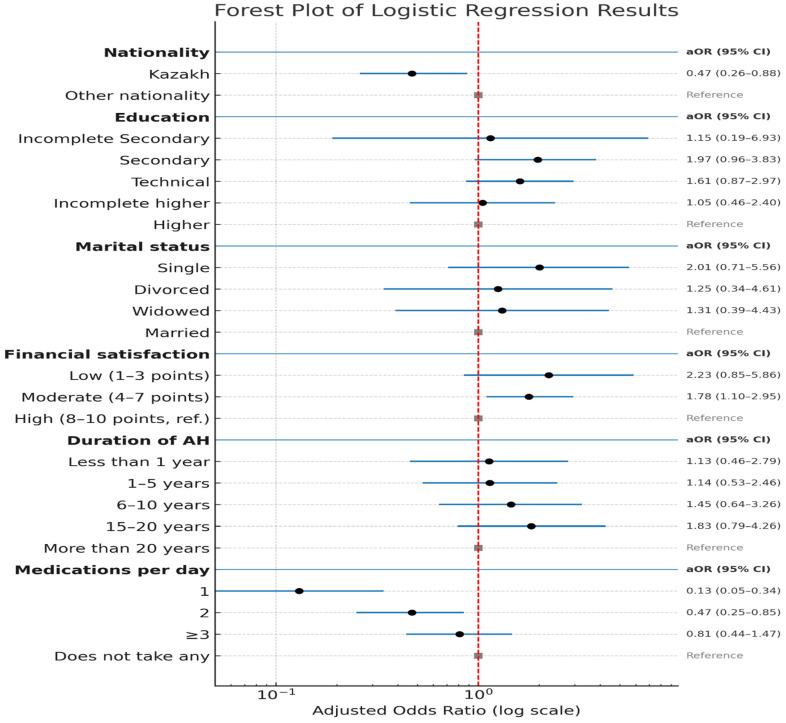
Results of multivariable logistic regression analysis. The red dashed line represents the MMAS-8 threshold score used to distinguish between high and low/medium medication adherence levels.

**Table 1 ijerph-22-01483-t001:** Demographic, socioeconomic and clinical characteristics of adherent and non-adherent groups.

Variables	Characteristic	Adherence, *N* (%)	Non-Adherence, *N* (%)	*p*
Age	M ± SD	58.7 ± 11.2	60.3 ± 10.6	0.24
Gender	Male	36 (37.5)	449 (43.6)	0.15
	Female	60 (62.5)	580 (56.4)	
Ethnicity	Kazakh	82 (85.4)	718 (69.8)	0.001
	Other ethnicities	14 (14.6)	311 (30.2)	
Education Level	Incomplete secondary	2 (2.2)	13 (1.3)	0.002
	Secondary	13 (13.5)	240 (23.3)	
	Technical	17 (17.7)	269 (26.1)	
	Incomplete higher	8 (8.3)	77 (7.5)	
	Higher	56 (58.3)	430 (41.8)	
Place of residence	Urban	90 (93.8)	934 (90.8)	0.03
	Rural	6 (6.3)	95 (9.2)	
Employment status	School student/University student	0	2 (0.2)	0.05
	Manual laborer/Worker	29 (30.2)	244 (23.7)	
	Technical staff	5 (5.2)	30 (2.9)	
	Government employee	3 (3.1)	74 (7.2)	
	Unemployed	4 (4.2)	27 (2.6)	
	Homemaker	11 (11.5)	72 (7.0)	
	Self-employed	2 (2.1)	57 (5.5)	
	Retired	41 (42.7)	508 (49.4)	
	Other	1 (1.0)	15 (1.5)	
Marital status	Single	7 (7.3)	46 (4.5)	0.003
	Married	70 (72.9)	799 (77.6)	
	Divorced	7 (7.3)	58 (5.6)	
	Widowed	12 (12.5)	126 (12.2)	
Duration of arterial hypertension	Less than 1 year	17 (17.7)	126 (12.3)	0.03
	1–5 years	36 (37.5)	349 (33.9)	
	6–10 years	16 (16.7)	229 (22.3)	
	15–20 years	12 (12.5)	198 (19.3)	
	More than 20 years	15 (15.6)	126 (12.3)	
Satisfaction with financial status	Low (1–3 points)	6 (6.3)	98 (9.5)	0.001
	Moderate (4–7 points)	51 (53.1)	690 (67.1)	
	High (8–10 points)	39 (40.6)	241 (23.4)	
Body Mass Index (BMI)	Overweight	38 (39.6)	444 (43.1)	0.02
	Normal	58 (60.4)	585 (56.9)	
Smoking status	Non-smoker	78 (81.2)	874 (84.9)	<0.001
	Smoker	18 (18.8)	155 (15.1)	
Comorbid conditions associated with hypertension	Present	60 (62.5)	717 (69.7)	0.04
	Absent	36 (37.5)	312 (30.3)	
Cardiovascular diseases	Present	32 (33.3)	287 (27.9)	0.03
	Absent	64 (66.7)	742 (72.1)	
Ischemic heart disease	Present	14 (14.6)	205 (19.9)	0.03
	Absent	82 (85.4)	824 (80.1)	
Angina pectoris	Present	6 (6.3)	104 (10.1)	0.04
	Absent	90 (93.8)	925 (89.9)	
Gastrointestinal disorders	Present	18 (18.8)	191 (18.6)	0.03
	Absent	78 (81.3)	838 (81.4)	
Neurological disorders	Present	3 (3.1)	75 (7.3)	0.05
	Absent	93 (96.9)	954 (92.7)	
Kidney diseases	Present	9 (9.4)	96 (9.3)	0.03
	Absent	87 (90.6)	931 (90.7)	
Eye diseases	Present	16 (16.7)	124 (12.1)	0.13
	Absent	80 (83.3)	905 (87.9)	
Number of prescribed medications per day	1	36 (37.5)	284 (27.6)	<0.001
	2	27 (28.1)	363 (35.3)	
	≥3	22 (22.9)	357 (34.7)	
	Does not take any	11 (11.5)	25 (2.4)	

## Data Availability

The datasets used and/or analyzed during the current study are available from the corresponding author upon reasonable request.

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
