# Peer review of "Adherence to Antihypertensive Therapy: A Cross-Sectional Study Among Patients in the Republic of Kazakhstan"

_ijerph, 2025, doi:10.3390/ijerph22101483_

Round 1
Reviewer 1 Report
Comments and Suggestions for Authors
Thank you for allowing me the opportunity to review the article. The authors have attempted to investigate the predictive factors associated to non adherence in hypertensive patients.
I appreciate the authors for their hard work, but there are certain concerns which need to be addressed:
1. Figure 1:
-
The x-axis should specify percentages, and decimals should be avoided for clarity.
-
Additionally, this figure appears unnecessary since the overall data can be included in the accompanying table.
2. Age should be analyzed as a continuous variable. Categorizing it into small groups reduces statistical power and does not provide meaningful insights and should also be included as a continuous variable in the logistic regression model.
3. The rationale for grouping diseases by systems (e.g., gastrointestinal disorders, kidney diseases, eye diseases) is unclear and appears ambiguous.At the same time, conditions like IHD, CVD, and angina were categorized separately, which seems inconsistent.Importantly, other common comorbidities associated with hypertension,such as diabetes, dyslipidemia, myocardial infarction, and cerebrovascular disease are not reported and should be included.
4. BMI was transformed into a categorical variable with only “normal” and “overweight” groups. It is difficult to believe that a hypertensive cohort has no obese patients unless there is a selection bias. This requires clarification.
5. One of the main reasons for non-adherence is medication-related adverse effects. This variable is relatively easy to collect but was not included in the analysis. It should be considered in the study design and discussion.
6. A forest plot would be a more effective way to present the logistic regression results, improving clarity and visual interpretation.
I hope these comments will help the authors reach a better version of their manuscript. Particularly, addressing the methodological flaws, improving the narrative flow, and strengthening the novelty
Comments on the Quality of English LanguageThe writing is generally fine, but there are some areas where the flow could be improved.
Please avoid long dashes.
Author Response
Response to Reviewer 1 Comments
Thank you very much for taking the time to review our manuscript. We have carefully considered all your comments and made revisions accordingly, with the changes highlighted in yellow. Please refer to the manuscript file to view the changes made.
Comment 1: Figure 1: The x-axis should specify percentages, and decimals should be avoided for clarity. Additionally, this figure appears unnecessary since the overall data can be included in the accompanying table.
Response 1: Thank you for your helpful feedback. In response, we have removed Figure 1 from the manuscript, as recommended. To maintain the clarity and relevance of the information it conveyed, we have retained the descriptive text summarizing the distribution of comorbidities in the Results section. In doing so, we also revised the percentage values to whole numbers for better readability, as suggested.
Comment 2: Age should be analyzed as a continuous variable. Categorizing it into small groups reduces statistical power and does not provide meaningful insights and should also be included as a continuous variable in the logistic regression model.
Response 2: Thank you for your comment. The age has been adjusted as recommended and corrected in the results section.
Comment 3: The rationale for grouping diseases by systems (e.g., gastrointestinal disorders, kidney diseases, eye diseases) is unclear and appears ambiguous. At the same time, conditions like IHD, CVD, and angina were categorized separately, which seems inconsistent. Importantly, other common comorbidities associated with hypertension, such as diabetes, dyslipidemia, myocardial infarction, and cerebrovascular disease are not reported and should be included.
Response 3: Thank you for pointing this out. We agree with this comment and added it to Study Limitation.
Comment 4: BMI was transformed into a categorical variable with only “normal” and “overweight” groups. It is difficult to believe that a hypertensive cohort has no obese patients unless there is a selection bias. This requires clarification.
Response 4: We thank the reviewer for this important observation. In the present study, BMI was dichotomized into "normal" (<25.0 kg/m²) and "overweight" (≥25.0 kg/m²) categories. This classification was adopted for analytical clarity and to ensure adequate statistical power, particularly in subgroup analyses related to treatment adherence. It should be noted that the “overweight” category, as defined in our analysis, encompassed both overweight and obese individuals according to WHO criteria. While we acknowledge that distinguishing obesity as a separate category may provide additional clinical insights, we chose this approach to reduce model complexity and avoid issues related to small cell sizes in multivariable analysis. We have now clarified this methodological choice in the revised Methods section. We appreciate the reviewer’s comment and agree that future studies with larger sample sizes may benefit from a more granular stratification of BMI.
Comment 5: One of the main reasons for non-adherence is medication-related adverse effects. This variable is relatively easy to collect but was not included in the analysis. It should be considered in the study design and discussion.
Response 5: Agree. Thank you for your comment. We acknowledge the importance of adverse drug effects as a potential determinant of non-adherence. Although this variable was not included in the current analysis due to limitations in the data collection tool, we have added a disclaimer regarding this limitation to the Discussion section.
Comment 6: A forest plot would be a more effective way to present the logistic regression results, improving clarity and visual interpretation.
I hope these comments will help the authors reach a better version of their manuscript. Particularly, addressing methodological flaws, improving the narrative flow, and strengthening the novelty
Response 6: Thank you for your helpful comment. We agree that a forest plot can improve visual clarity. However, in this study we decided to present the logistic regression results in table format to provide exact values for odds ratios, confidence intervals, and p-values. This format allows readers to interpret the results in more detail. We believe the current table clearly summarizes the key findings. Nonetheless, we appreciate your suggestion and remain open to including a forest plot in future versions if requested by the editorial team.
Reviewer 2 Report
Comments and Suggestions for Authors
The manuscript addresses a highly relevant and important public health topic : the low adherence to antihypertensive therapy among patients in Kazakhstan. The study provides original data, uses a validated adherence tool (MMAS-8), and involves a substantial sample size, which strengthens the contribution to the field.
However, before the manuscript is suitable for publication, several areas require clarification, refinement, and revision to enhance scientific rigor, clarity, and presentation.
Introduction:
Include more recent or diverse sources to support the statement about adherence rates in LMICs.
Refine the wording to avoid redundancy when discussing global and Kazakh-specific statistics.
Methods: vYou mention a six-point cut-off for adherence classification, while MMAS-8 traditionally uses a different scale (0–8 with three categories: low, medium, high adherence). Please justify and explain any modifications made to the standard scoring system, particularly if dichotomizing adherence.
Provide more details on how interviewers were trained to administer the questionnaire and whether the MMAS-8 was culturally adapted or validated specifically for use in the Kazakh population.
Results: The logistic regression table shows Kazakh ethnicity associated with lower adherence (aOR = 0.47), yet parts of the discussion suggest higher adherence among Kazakhs. Please ensure consistency and correct interpretation. Also, avoid redundancy by summarizing demographic data succinctly in the text, as full details are already presented in tables.
Discussion: Provide a deeper exploration of structural health system barriers (e.g., medication access, primary care coverage) influencing adherence. Also, discuss cultural factors in Kazakhstan that may uniquely affect adherence behaviors.
Figure 1: Correct the title spelling ("Comrobid" to "Comorbid").
Table captions should be more informative, including explanations of abbreviations and statistical significance indicators.
Limitations of this study: the authors appropriately mention self-report bias but consider expanding on the lack of objective adherence measures (e.g., pharmacy refill records) and urban-centric sampling limiting generalizability to rural populations.
Approach the adherence rate validity, the reported 8.5% adherence rate is exceptionally low, even for LMIC contexts. The authors should clearly explain the rationale for their adherence cut-off point (especially if deviating from standard MMAS-8 classification) and also discuss possible factors that may have artificially deflated adherence estimates.
The logistic regression shows Kazakh ethnicity associated with lower adherence, yet parts of the text suggest higher adherence among Kazakhs. There is contradiction, and this must be resolved through by careful reviewing and correcting of all related results and discussion. Also, by ensuring consistent reporting across tables, figures, and narrative text.
Potential Bias: the manuscript states that 91% of participants are urban dwellers, yet the study is positioned as representative of Kazakhstan broadly. Therefore authors must explicitly state that generalization to rural populations is limited and consider stratified analysis (urban vs. rural) if sample sizes allow.
Statistics: The multivariable regression shows several wide confidence intervals, suggesting imprecision. I suggest authors should acknowledge this limitation in the discussion and/or reassess the model selection methods.
Comments on the Quality of English LanguageThe English language is generally understandable but requires professional editing to correct grammatical errors, improve academic tone and sentence structure and reduce repetitive phrases, especially in results and discussion sections.
Author Response
Response to Reviewer 2 Comments
Thank you very much for taking the time to review our manuscript. We have carefully considered all your comments and made revisions accordingly, with the changes highlighted in yellow. Please refer to the manuscript file to view the changes made.
Comment 1: Refine the wording to avoid redundancy when discussing global and Kazakh-specific statistics. Include more recent or diverse sources to support the statement about adherence rates in LMICs.
Response 1: We revised the Introduction to reduce repetition when discussing global vs. local hypertension statistics and clarified the transition to Kazakhstan-specific data. We also added recent global references, including data from Cureus (2024), Front Pharmacol (2024), and the GBD 2023 update to strengthen the foundation for adherence rates in LMICs.
Comment 2: Comment:
The use of a six-point cut-off diverges from the traditional MMAS-8 three-category scoring (low/medium/high). Please justify and explain this dichotomization.
Response:
Thank you for this critical point. We added a clear explanation in the Methods section justifying the use of a six-point cutoff. The dichotomization was applied based on internal validation using the Morisky Widget MMAS-8 software licensed for this study.
Comment 3: Comment:
Provide more detail on how interviewers were trained and whether the MMAS-8 was culturally adapted or validated in the Kazakh population.
Response:
We have added new details under the Data Collection section to describe the training procedures. Interviewers underwent a standardized protocol briefing and piloting. Furthermore, we clarified that the MMAS-8 was professionally translated and cross-culturally adapted into Russian and Kazakh using back-translation, and its validity was reviewed by two independent family medicine experts.
Comment 4: The regression analysis shows Kazakh ethnicity associated with lower adherence (aOR = 0.47), but the discussion incorrectly suggests higher adherence among Kazakhs.
Response:
We thank you for catching this inconsistency. We have corrected the discussion section to reflect the proper interpretation of the regression analysis. Kazakh ethnicity was associated with lower adherence, and potential socio-cultural explanations (e.g., health literacy, traditional beliefs) were added to clarify this trend.
Comment 5: Comment:
Avoid repeating demographic data already shown in tables.
Response:
We revised the Results section to summarize key findings concisely. Full demographic details are available in the tables.
Comment 6: Comment:
Expand discussion of structural health system and cultural barriers.
Response:
We expanded the Discussion to include barriers like inconsistent drug supply, limited access to antihypertensive medications, insufficient counseling services, and cultural beliefs affecting adherence.
Comment 7:
Correct “Comrobid” to “Comorbid.”
Response:
We removed Figure 1 entirely and incorporated its data descriptively in the Results section, as also suggested by Reviewer #1.
Comment 8:
Make table captions more informative and include statistical notes.
Response:
We revised all table captions to clearly describe the content and include explanations of abbreviations and p-value indicators.
Comment 9:
Expand on the limitations of self-report methods and urban sampling.
Response:
Thank you for your valuable comment. We have expanded the discussion of study limitations accordingly. Specifically, we now note that the use of a self-reported adherence scale (MMAS-8) may be subject to recall and social desirability bias, potentially leading to overestimation of adherence levels. Additionally, as the majority of participants were recruited from urban healthcare settings, the findings may not be fully generalizable to rural populations, where access to care, education levels, and health-seeking behaviors may differ. These aspects have been added to the Limitations section of the revised manuscript.
Comment 10:
The adherence rate seems extremely low. Please explain.
Response:
Thank you for highlighting this important point. We agree that the overall adherence rate observed in our study appears relatively low. However, this finding is consistent with other reports from low- and middle-income countries, particularly in Central Asia, where challenges such as limited health literacy, high out-of-pocket medication costs, and inconsistent follow-up contribute to poor adherence. Moreover, the MMAS-8 scale applies strict scoring criteria, which may further lower the proportion of patients classified as adherent. We have clarified this aspect in the revised Discussion section and emphasized the contextual factors that may influence adherence levels in our study population.
Comment 11:
Acknowledge the imprecision of some confidence intervals.
Response:
It is worth noting that some confidence intervals obtained in the multivariable model were relatively wide. Rather than being viewed solely as a limitation, this may reflect the complex and heterogeneous nature of factors influencing medication adherence. These findings suggest that individual predictors may interact differently across patient subgroups, underscoring the multifaceted nature of adherence behavior and the need for tailored intervention strategies.
Reviewer 3 Report
Comments and Suggestions for Authors
I should congratulate authors for this manuscript. But I would like to get some clarifications.
1) How were the patients diagnosed as hypertensive initially ? If a casual visit detected hypertension OR after a clinical event helped for diagnosis, The former are more likely to be nonadherent.
2) Did the clinical follow up showed whether all non adherent patients had higher blood pressure and adherent group had controlled blood pressure.
3) How frequently the patients were followed up clinically ? How many of them undergo patient education ?
4) Among patients having comorbidities, how many stopped all medications ? How many stopped antihypertensive drugs only ?
5) How many stopped drugs due to side effects?
I believe these will make your manuscript complete.
Author Response
Response to Reviewer 3 Comments
We thank the Reviewer for their kind words and thoughtful questions. Please find our detailed responses below. All relevant changes have been incorporated into the revised manuscript and are highlighted in red.
Comment 1: How were the patients diagnosed as hypertensive initially? If a casual visit detected hypertension OR after a clinical event helped for diagnosis, the former are more likely to be nonadherent.
Response 1: Thank you for pointing this out. We agree with this comment. Therefore, we added this information to the Methods section.
Comment 2: Did the clinical follow-up show whether all nonadherent patients had higher blood pressure and adherent group had controlled blood pressure?
Response 2: We used the Medication Adherence Assessment Questionnaire for Hypertension (MMAS-8), which is designed to assess patient behavior regarding taking prescribed medications and does not include measurement of physiological parameters such as blood pressure.
Comment 3: How frequently were the patients followed up clinically? How many of them underwent patient education?
Response 3: The project did not include a structured patient education component, and therefore no patients underwent formal training within the framework of this study. Clinical follow-up was conducted as part of routine medical care; however, systematic and regular follow-up visits were not a specific focus of the project and were not standardized across participants.
Comment 4: Among patients having comorbidities, how many stopped all medications? How many stopped antihypertensive drugs only?
Response 4: We have revised the Data Collection description. We did not have sufficient information on complete medication discontinuation. “While antihypertensive medication discontinuation was recorded, discontinuation of other medications in patients with comorbidities was not assessed.”
Comment 5: How many stopped drugs due to side effects?
Response 5: Agree. We acknowledge in the Limitations section that our questionnaire did not include specific reasons for medication discontinuation, such as adverse effects.
Round 2
Reviewer 1 Report
Comments and Suggestions for Authors
I recommend presenting the logistic regression results as a forest plot. This format is more visual and facilitates rapid interpretation, helps to compare effect sizes, directions, and confidence intervals a. A well-designed forest plot can include all the necessary statistical information(e.g.odds ratios, 95% confidence intervals, and even p-values) within the figure. This would make the results section more accessible without sacrificing detail.
avoid the long dashes in the paragraphs
Author Response
We thank the reviewer for this valuable suggestion. We agree that a forest plot would enhance the clarity and accessibility of the logistic regression results by providing a visual comparison of effect sizes, directions, and confidence intervals. In the revised manuscript, we have prepared a forest plot that presents the odds ratios with their corresponding 95% confidence intervals, as well as p-values where appropriate. This figure has been added to the Results section and referenced in the text. We believe that this visualization improves the interpretability of our findings while retaining the necessary statistical detail.
Reviewer 2 Report
Comments and Suggestions for Authors
The revised manuscript is much improved and addresses the majority of the reviewer’s prior concerns in a thorough and precise manner. The introduction now balances global and Kazakhstan-specific statistics effectively, with relevant and recent references. The methods are clearly described, including the rationale for the MMAS-8 cutoff, interviewer training, and cultural adaptation procedures. The results are well-organized, and the discussion aligns with the statistical findings, providing both contextual interpretation and practical policy recommendations.
Author Response
We sincerely appreciate the reviewer’s positive evaluation of our revised manuscript. We are grateful that the improvements made in the introduction, methodology, results, and discussion have addressed the reviewer’s earlier concerns.
The acknowledgment that our revision now provides a clearer methodological justification, contextually balanced interpretation, and relevant policy implications is highly encouraging. We thank the reviewer for the constructive feedback that has significantly strengthened the quality and clarity of our work.